# Tumor Characteristics and Treatment Responsiveness in Pembrolizumab-Treated Non-Small Cell Lung Carcinoma

**DOI:** 10.3390/cancers16040744

**Published:** 2024-02-10

**Authors:** Haiyan Li, Sunitha Shyam Sunder, Karan Jatwani, Yongho Bae, Lei Deng, Qian Liu, Grace K. Dy, Saraswati Pokharel

**Affiliations:** 1Department of Pathology and Laboratory Medicine, Roswell Park Comprehensive Cancer Center, Buffalo, NY 14203, USA; hyl1516@gmail.com (H.L.); sunitha.shyamsunder@roswellpark.org (S.S.S.); 2Department of Medicine, Roswell Park Comprehensive Cancer Center, Buffalo, NY 14203, USA; karan.jatwani@roswellpark.org (K.J.); ldeng1@fredhutch.org (L.D.); grace.dy@roswellpark.org (G.K.D.); 3Department of Pathology and Anatomical Sciences, University at Buffalo, Buffalo, NY 14203, USA; yonghoba@buffalo.edu; 4Department of Biostatistics and Bioinformatics, Roswell Park Comprehensive Cancer Center, Buffalo, NY 14203, USA; qian.liu@roswellpark.org

**Keywords:** pembrolizumab, non-small cell lung cancer, tumor pathological features, RECIST, tumor-infiltrating lymphocytes

## Abstract

**Simple Summary:**

Non-small cell lung cancer accounts for most lung cancer. In recent years, immunotherapy, especially with PD1/PDL1 inhibitors such as pembrolizumab, has emerged as an important treatment modality. The exploration of new avenues such as tumor biomarker analysis has proven beneficial in determining treatment responsiveness. Various studies have shown a positive association between distinct tumor characteristics and the response to pembrolizumab. In this study, we established that the presence of higher proportions of viable tumor cells and increased tumor-infiltrating lymphocytes in non-small cell lung cancer show favorable responses to treatment with pembrolizumab.

**Abstract:**

Pembrolizumab, a widely used immune checkpoint inhibitor (ICI), has revolutionized the treatment of non-small cell lung cancer (NSCLC). Identifying unique tumor characteristics in patients likely to respond to pembrolizumab could help the clinical adjudication and development of a personalized therapeutic strategy. In this retrospective study, we reviewed the clinical data and pathological features of 84 NSCLC patients treated with pembrolizumab. We examined the correlation between the clinical and demographic characteristics and the tumor histopathologic features obtained before immunotherapy. The response to pembrolizumab therapy was evaluated via the Response Evaluation Criteria in Solid Tumors (RECIST). The clinical data and cancer tissue characteristics were assessed and compared among three groups according to the following RECIST: the responsive group (RG), the stable disease group (SD), and the progressive disease group (PD), where the RG comprised patients with either a complete response (CR) or a partial response (PR). The overall survival rate of the RG group was significantly higher than the SD and PD groups. In addition, the percentage of pre-treatment viable tumor cell content in the RG and SD groups was significantly higher. At the same time, the extracellular stroma proportion was significantly lower than that of the PD group. The number of tumor-infiltrating lymphocytes (TILs) in the RG group was significantly higher than in the PD group. There were no significant differences in tumor necrosis, the stroma composition, PD-L1 expression level (TPS 1–49% vs. ≥50%), and treatment response. In conclusion, our population of NSCLC patients who experienced positive treatment responses to pembrolizumab therapy had a better prognosis compared to patients with either SD or PD. Moreover, the relative proportions of viable tumor cells to tumor-associated lymphocytes were associated with responsiveness to treatment. It is expected that larger prospective clinical studies will further validate these findings.

## 1. Introduction

Lung cancer is the leading cause of cancer-related deaths in the United States, with a mortality rate of 35 per 100,000 of the population. Of all these cases, 13% are attributed to small cell lung cancer and 84% to non-small cell lung cancer (NSCLC) [1]. The majority of cases are diagnosed at an advanced stage, resulting in a grim prognosis [2]. Various chemotherapeutic systemic therapy options, including platinum-based drugs and targeted therapies, have been used over time [3]. In recent years, the immunotherapeutic modality has emerged as a promising approach at various stages of lung cancer. This includes immune checkpoint inhibitors (ICIs), other non-specific immune modulators (toll-like receptor agonists, etc.), and cancer vaccines [4]. With regard to ICIs, the modulation of cytotoxic T-lymphocyte-associated protein 4 (CTLA-4), programmed death cell protein 1 (PD-1)/programmed death ligand 1 (PD-L1), LAG3, B7-H3, and B7-H4 on CD8+ tumor-infiltrating lymphocytes (TILs) are the primary targets [5].

PD-L1, also known as Cluster of Differentiation 274 (CD274), is expressed in numerous tumor cells, including NSCLC and antigen-presenting cells (APCs), and is one of the primary ligands of the PD-1 receptor [6]. PD-1 is a member of the immunoglobulin gene superfamily and is prominently expressed in T cells. Ligating tumor-associated PD-L1 with PD-1 promotes tumor progression through reduced T-cell proliferation and function [7]. Inhibition of the PD-L1/PD-1 axis is the foundation for most of the anti-cancer effects of ICIs in the clinic. Multiple NSCLC clinical trials such as KEYNOTE-189, IMpower150, IMpower010, KEYNOTE-091, CheckMate 816, and PACIFIC have shown extended survival rates in patients treated with ICIs [8,9,10,11,12,13]. Examples of anti-PD-1 antibodies include nivolumab, pembrolizumab, and cemiplimab, and those against PD-L1 include atezolizumab, durvalumab, and avelumab. Pembrolizumab, a humanized monoclonal immunoglobulin IgG4 antibody targeting PD1 and blocking its interaction with PD-L1, was the first ICI to receive US FDA approval in 2016 for first-line treatment for advanced non-small cell lung cancer (NSCLC) in PD-L1 high EGFR/ALK wildtype NSCLC [14]. However, despite biomarker selection, ICIs, including pembrolizumab, are only effective in a subset of patients [15]. The factors affecting the efficacy of ICI therapy are the subject of intense investigation.

TILs and cytotoxic T lymphocytes in the tumor stroma have been linked to improved survival rates in NSCLC and a positive response to ICI therapy, respectively [16]. In addition, TILs’ immune gene expression signatures are touted as predictors of durable benefits from ICIs [17]. Previous studies have reported that biomarkers such as including stromal CD4 TILs, high PD-L1 expression, and intratumoral CD8 TILs are predictors of OS and progression-free survival in NSCLC [18]. However, the results reported in the prior studies are not uniform.

In this retrospective study involving 84 NSCLC patients treated with pembrolizumab, we explore the correlation between clinical/demographic features and pre-immunotherapy tumor histopathologic characteristics. Using tumor samples obtained before pembrolizumab treatment, we specifically examined tumor cellularity, tumor necrosis, tumor stroma proportions, TILs, tumor-infiltrating T cells, PD-L1 expression, and molecular profile. Subsequently, we compared these features with the treatment responsiveness in patients with advanced-stage NSCLC treated with pembrolizumab.

## 2. Materials and Methods

### 2.1. Patients and Tissue Samples

The Roswell Park Cancer Institute Institutional Review Board (IRB) approved this retrospective project in compliance with federal, state, and local requirements. All clinical and outcome patient data were collected. Tumor specimens were collected from patients who underwent biopsy for lung cancer at Roswell Park Cancer Institute (RPCI), Buffalo, NY, USA. We reviewed the clinical data of 84 patients with stage III and IV NSCLC who had been treated with pembrolizumab monotherapy from October 2016 to June 2021. The clinical information of all the patients, including their age at diagnosis, gender, clinical staging, history of tobacco abuse, previous medical history, treatment, and response, were retrieved from the electronic medical records (EMR).

For each eligible patient, we acquired a suitable archival tissue sample containing an adequate amount of tumor tissue (specifically, containing at least 100 tumor cells). The tissue collection took place before the commencement of treatment with pembrolizumab. Pathological features were extracted from 43 tissue cases wherein archival tissue was available in our tissue bank. All tissue analyses were conducted at the Roswell Park Comprehensive Cancer Center, Department of Pathology.

### 2.2. Assessment of Tumor Histology and Molecular Studies

The pathological features, including tumor cell content (%), necrosis (%), the ratio of TILs to tumor cells (TC), and stromal component (%), were examined microscopically on hematoxylin and eosin (HE)-stained slides of either a core biopsy or surgical resection specimen that were used to diagnose patients with NSCLC at our institute. The patients’ molecular and cytogenetics features were analyzed by next-generation sequencing (NGS)-based comprehensive genomic profiling. RNA sequencing was also performed in selected cases to examine the gene expression profiling (GEP) of the tumor microenvironment (TME).

### 2.3. Tumor PD-L1 Scoring

The expression of PD-L1 was examined by IHC staining using an antibody (PD-L1 IHC 22C3 pharmDx) kit (Agilent, Santa Clara, Ca, USA). Immunostaining was performed following the manufacturer’s protocol. PD-L1 protein expression was determined using the Tumor Proportion Score (TPS) described before [19], which includes the percentage of viable tumor cells showing partial or complete membrane staining at any intensity. An assessment of expression levels was performed in sections that included at least 100 viable tumor cells. The specimen was considered to have low PD-L1 expression if TPS ≥ 1% and high PD-L1 expression if TPS ≥ 50%. The PD-L1 expression status was evaluated by experienced thoracic pathologists and blinded for treatment outcome.

### 2.4. Efficacy Assessment

The response to pembrolizumab therapy was evaluated by the Response Evaluation Criteria in Solid Tumors version 1.1 (RECIST 1.1) [20]. The clinical data and cancer tissue characteristics were assessed and compared among three groups according to the following RECIST criteria: complete response (CR) or partial response (PR) as a responsive group (RG), stable disease group (SD), and progressive disease group (PD).

### 2.5. Statistical Analysis

The clinical demographic characteristics were summarized and presented by response groups (RG, SD, PD) using summary statistics such as median and IQR for continuous variables, and frequencies and contingency tables for categorical variables. The difference in each clinical variable between the three response groups was tested using appropriate statistical methods based on the type and distribution of data. The Kruskal–Wallis rank sum test was conducted for continuous variables. Pearson’s Chi-square test or Fisher’s exact test was conducted for categorical variables. For survival outcomes, Kaplan–Meier survival curves were generated and the log-rank test was conducted to compare different response groups. Other clinicopathological features were tested using independent two-sample T-tests or one-way ANOVA (Analysis of variance) to measure the differences between two or more groups. A *p*-value of <0.05 was considered statistically significant. Data analysis was performed using the R programming language version 4.2.2 and GraphPad Prism version 10.0.2 (232) (GraphPad Software Inc., La Jolla, CA, USA).

## 3. Results

The 84 cases of NSCLC that underwent treatment with pembrolizumab from October 2016 to June 2021 at Roswell Park Comprehensive Cancer Center were included in this study.

### 3.1. Patient Characteristics

Of the patients included in this study, 48 were female (57.1%) and 36 were male (42.9%). The median age was 67, with ages ranging from 43 to 89. Fifty patients were diagnosed with adenocarcinoma (59.5%), 18 patients were diagnosed with squamous cell carcinoma (21.4%), and 1 patient was diagnosed with adenosquamous carcinoma. In comparison, 15 patients were diagnosed with non-small cell carcinoma, not otherwise specified (17.9%) at the time of initial diagnosis. History of tobacco use was also examined for all patients: 28 patients (33.3%) were current smokers, 48 patients (54.8%) were former smokers, and 8 patients (9.5%) had never smoked in their lives. By June 2021, 68 patients (81.0%) had discontinued pembrolizumab therapy, while 16 patients (19.0%) were still undergoing treatment. The reasons for the discontinuation of therapy included excellent response in 7 patients (10.3%), completion of the two-year treatment for 3 patients (4.4%), adverse events in 16 patients (23.5%), disease progression in 30 patients (44.1%), frailty in 6 patients (8.8%), the start of chemotherapy in 1 patient (1.5%) and, tyrosine kinase inhibitor (TKI) therapy in 1 patient (1.5%). One patient (1.5%) was lost to follow-up. In total, four patients (4.8%) showed a completed response (CR) to the pembrolizumab treatment, while 38 patients (45.2%) showed a partial response (PR). Also, 23 patients (27.4%) showed stable disease (SD) and 19 patients (22.6%) showed progressive disease (PD) at the time of follow-up evaluation (Table 1).

### 3.2. Treatment Outcome

The overall survival of all patients who received pembrolizumab treatment was compared for the following three groups based on their treatment responses: the Response group (RP) with 42 patients, the Stable Disease group (SD) with 23 patients, and the Progressive Disease group (PD) with 19 patients.

As of April 2023, 26 patients (61.9%) in the Response survival group remained alive, and the median survival for this cohort was 42 months. In the SD group, 23 patients (39.9%) remained alive, with a median survival in this cohort of 24 months. In the PD group, 17 patients (89.4%) were deceased, with a median survival of 5 months (Figure 1).

Patients who responded to pembrolizumab treatment demonstrated a significantly higher survival rate compared to patients with stable disease or progressive disease (*p* < 0.0001). Additionally, the survival rate of patients with stable disease was also significantly higher than that of patients with progressive disease (*p* < 0.0001). The Kaplan–Meier survival curves are presented in Figure 1.

### 3.3. Tumor-Infiltrating Lymphocytes and CD8+ Lymphocytes

The levels for CD8 TILs were recorded from the molecular profiling data (OmniSeq Inc., Buffalo, NY, USA) obtained from patients’ medical records. RNA sequencing was employed to analyze CD8+ cytotoxic T-lymphocytes. The interpretation of gene expression for CD8 TIL was performed as follows: each gene was compared to a reference population derived from 735 unique tumors, normalized to a value of between 1 and 100, and scored as the percentile (relative) rank (%rank). The gene expression of CD8+ TILs was expressed as high for genes ranked 75–100 (highly inflamed), moderate for genes ranked 25–74 (moderately inflamed), and low for genes ranked 0–24 (non-inflamed). The results were stratified as “inflamed” if CD8+ lymphocytes were observed in high or moderate levels, and as “non-inflamed” if CD8+ lymphocytes were present in low quantities as per the report. The tumor was reported as inflamed in 46 patients (54.8%) and non-inflamed in 38 patients (45.2%). Of the 46 patients with a high level of TILs by gene expression, 29 demonstrated either a complete or partial response to pembrolizumab treatment. In comparison, there were 17 non-responders. This difference was statistically significant (*p* < 0.05). Of the 38 patients with a non-inflamed status, 13 patients showed either a complete or partial response to the pembrolizumab treatment and 25 were non-responders (Figure 2A).

Additionally, we conducted a comparison of the tumor-infiltrating lymphocytes (TILs) on HE-stained slides for the different groups previously classified based on their response to pembrolizumab therapy. The average number of TILs was 9.1 ± 1.5/HPF in the RG, 10.5 ± 4.0/HPF in the SD group, and 4.5 ± 1.6/HPF in the PD group. The number of TILs per high-power field in the RG was notably higher than that in the PD group (*p* < 0.05) (Figure 2B). We further calculated the ratio of TIL to tumor cells (TC), which represents the number of TILs within 100 tumor cells per high-power field (400× magnification). This ratio ranged from 0.1 to 7.1. Subsequently, we categorized the samples into two groups: the low infiltrate ratio group (ratio < 1) and the high infiltrate ratio group (ratio ≥ 1). Of the 16 patients who exhibited a response to treatment (RG), 3 patients displayed a low lymphocyte infiltration ratio, while the remaining 13 had a high lymphocyte infiltration ratio. In contrast, of the six patients who experienced disease progression (PD), four exhibited a low lymphocyte infiltration ratio, and two had a high lymphocyte infiltration ratio (*p* < 0.05) (Figure 2C).

### 3.4. PD-L1 Expression Status

All patients showed PD-L1 expression on the tumor cells. The average TPS in all 84 patients was 62.7%. In the RG group, 7 patients had low PD-L1 expression with an average TPS of 10.4%, and 35 patients had high PD-L1 expression with an average TPS of 72.1%. In the SD group, 5 patients had low PD-L1 expression with an average of 9.2%, and 17 patients had high PD-L1 expression with an average TPS of 70.7%. In the PD group, 2 patients had low PD-L1 expression with an average of 10.5%, and 15 patients had high PD-L1 expression with an average TPS of 79.3% (Figure 3). There was no significant difference in the PD-L1 expression status on tumor cells among these three groups (*p* > 0.05).

### 3.5. Somatic Molecular Alterations

Molecular studies were performed in formalin-fixed paraffin-embedded (FFPE) samples, mostly at Omniseq Comprehensive. At least one gene mutation was detected in 79 of the 84 patients (94.0%), whereas no gene mutation was detected in 5 patients. A total of 42 patients (50.0%) had a TP53 mutation, and 38 patients had a KRAS mutation. A total of 12 patients (14.3%) had both TP53 and KRAS gene mutations. Mutations were also detected in EGFR, BRAF, CDKN2A, RAF1, STK11, ATM, IDH, MARP2K, PIK3CA, PTEN, and FGFR. In addition, there were some other rare gene mutations. There was no significant difference among the three response groups (*p* > 0.05) in terms of gene mutations (Table 2). Additional data are provided in Appendix A.

### 3.6. Histopathological Assessment of Tumor Characteristics including Tumor Cellularity and Tumor-Infiltrating Lymphocytes

The overall tumor characteristics were assessed microscopically in hematoxylin-eosin-stained whole-tissue sections, including core biopsies and surgically resected specimens when available. Archival slides were available for review from a total of 33 unique patient cases, including core biopsies and surgical resections from patients who underwent procedures at Roswell Park Comprehensive Cancer Center. Of these patients, 16 were from the RG, 11 from the SD, and 6 from the PD groups.

In the RG group, the average proportion of tumor cells constituted 70.9 ± 4.0%, with extracellular stroma comprising 23.1 ± 3.8%, and necrotic tissue accounting for 5.9 ± 3.2% of the total lesional tissue. Conversely, in the SD group, the average tumor area occupied 66.8 ± 8.2% of the lesion, while the extracellular stroma comprised 28.2 ± 6.8% of the total, and necrotic tissue was present in 5.0 ± 4.5% of the lesion. In the PD group, the average tumor cell area represented 48.3 ± 10.8% of the total lesion, with extracellular stroma comprising 41.7 ± 13.0% and necrotic tissue making up 10.0 ± 6.8%. The average tumor cell proportion in patients responding to pembrolizumab treatment was significantly higher than in patients experiencing disease progression (*p* < 0.05) (Figure 4A,B and Figure 5A). However, there was no significant difference observed between the response group and the stable disease group (*p* > 0.05) (Figure 5). Similarly, there were no significant differences in the proportion of tumor stroma or necrosis between different groups (Figure 4C,D and Figure 5B,C). 

## 4. Discussion

In our investigation, we analyzed the interrelations among the clinical and demographic parameters, histopathological features, and molecular characteristics of tumor specimens concerning the therapeutic efficacy of pembrolizumab in 84 NSCLC patients undergoing treatment. Our findings indicate that patients manifesting a favorable response to pembrolizumab treatment exhibit distinct tumor characteristics compared to non-responsive patients. These distinctive pathological features include the relatively high proportion of viable tumor cells and the higher number of tumor-infiltrating lymphocytes, both of which are correlated with therapeutic responsiveness. To the best of our knowledge, our study represents the first report showing the strong relation between the proportion of viable tumor cells and treatment responsiveness following pembrolizumab therapy in NSCLCs.

This is the first study to report the relationship between the percentage of viable tumor cells and treatment responsiveness after checkpoint inhibitor therapy. Previous studies have examined the tumor-to-stroma ratio and patient survival after surgery. NSCLCs with stroma-rich tumors were shown to have a poorer prognosis after surgical resection in some reports [21,22]. The relationship between tumor stroma and treatment outcome had not been examined before. We noted that a significantly higher proportion of viable tumors was present in the tissue samples from the patients responding to pembrolizumab treatment than in those showing disease progression. However, we did not see a significant association between the tumor stromal percentage or necrosis and treatment responsiveness in this study. Larger prospective clinical studies will be needed to validate these findings fully.

Several reports have explored the correlation between *TILs* and prognosis in NSCLC patients. A pooled analysis of 23 studies showed that the CD8+ T cells present within the tumor and stromal compartment had a better prognosis in terms of overall and disease-specific survival [23]. Similarly, other studies have reported the prognostic significance of stromal *TILs* in NSCLC patients as powerful predictors of PD-1 blockade [24]. Stromal CD8+ TILs are the strongest predictors of progression-free survival and overall survival [25]. Additionally, stromal CD4+ TILs have also been identified as viable biomarkers for predicting outcomes in some reports [18]. In addition to T-cells, various immune cells within the tumor microenvironment have also been linked to disease prognosis [26]. According to a meta-analysis published by Soo et al., the presence of dendritic cells, natural killer cells, M1 macrophages, CD8+ T cells, and B cells in both tumor and stroma areas is linked to a better prognosis for NSCLCs. In contrast, the presence of stromal M2 macrophages, regulatory T cells, and PD-L1 overexpression is linked to a poorer prognosis [23]. It is important to note that our study specifically focused on TILs and CD8+ cells. Expanding the research to encompass a broader array of immune cell markers would provide a valuable corroboration of these prior findings. Furthermore, our investigation revealed that the lymphocyte-to-tumor cell ratio between the responsive and disease progression groups aligns consistently with the total lymphocyte counts. TILs are distributed heterogeneously, and their predictive value may be diminished when an assessment is made based on a small tissue sample. Our observation suggests that in cases of small biopsies, the TIL-to-tumor cell ratio may serve as a potential predictor of the therapeutic response to pembrolizumab in NSCLC patients.

The utility of PD-L1 expression as a biomarker for predicting the effectiveness of immune checkpoint inhibitors has been a topic of debate. Previous trials involving nivolumab have yielded conflicting results when examining the relationship between PD-L1 IHC and key outcome parameters such as the overall response rate (ORR), overall survival (OS), and progression-free survival (PFS) (PMID: 28636851) [27]. In another study, in patients with advanced NSCLC and PD-L1 expression, pembrolizumab was associated with significantly longer progression-free and overall survival and with fewer adverse events in comparison to platinum-based chemotherapy [28]. Similarly, the correlation between PD-L1 expression levels and treatment responsiveness after pembrolizumab was reported in another study [29]. In our study, we did not identify any significant association between tumor PD-L1 expression as a standalone marker and a durable response. One plausible explanation for these disparities could be the variations in the PD-L1 scoring scale used in the earlier study, which might lead to differences in the interpreted results. The estimation and reporting methods used for PD-L1 have varied in different studies in the literature [30]. Despite the presence of biological variations in these indicators for different cancers, there is a need for the standardization of these parameters and optimal cutoffs to function as reproducible and reliable indicators for ICI response [31].

This study’s retrospective nature and limited sample size were notable constraints. Additionally, including patients from across a spectrum of disease stages, from stage II to IV, introduces potential variability in the outcome data. Additionally, the study’s sample included both biopsies and resection specimens, but a predominant portion was constituted by biopsy tissue. This bias stems from the infrequent use of surgical resection in treating patients with advanced disease stages. To substantiate these findings, prospective studies investigating tumor characteristics in a more uniform cohort of patients treated with immunotherapy are warranted.

## 5. Conclusions

To summarize, our study revealed that NSCLC patients undergoing pembrolizumab treatment with favorable responses exhibited specific tumor characteristics including higher proportions of viable tumor cells and increased tumor-infiltrating lymphocytes, indicating treatment effectiveness. Notably, our study is the first to establish a strong connection between viable tumor cell proportions and pembrolizumab response in NSCLCs.

## Figures and Tables

**Figure 1 cancers-16-00744-f001:**
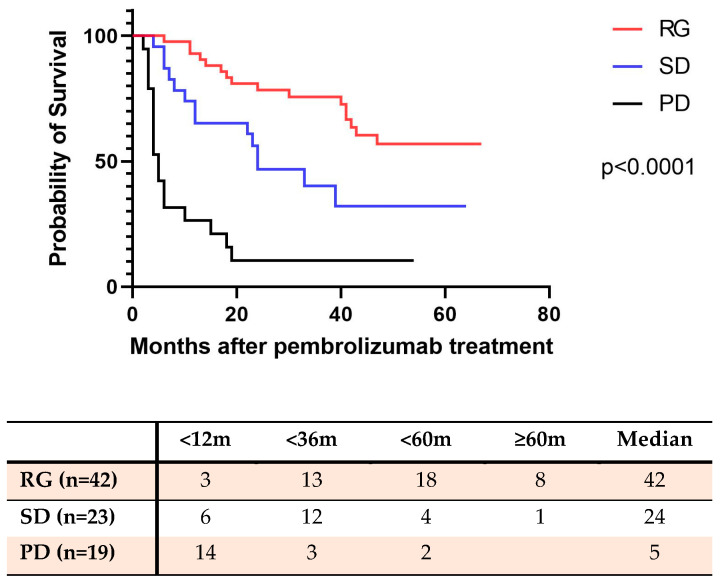
The overall survival rate of NSCLC patients with pembrolizumab therapy (as of April 2023). RG: Response group (complete response plus partial response); SD: Stable disease group; PD: Progressive disease group.

**Figure 2 cancers-16-00744-f002:**
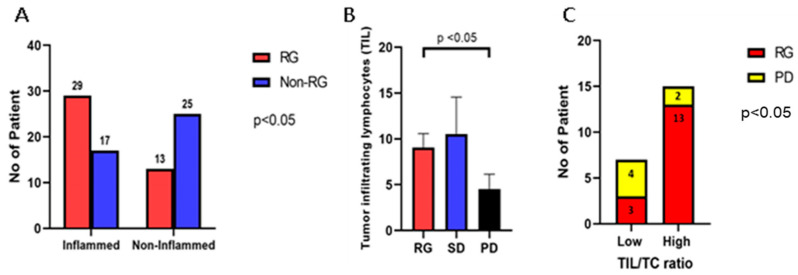
CD8^+^ T-cell, tumor-infiltrating lymphocytes (TILs), and the TIL to tumor cell (TC) ratio according to treatment-responsive status. (**A**) In the treatment-responsive group (complete and partial response), 29 of the 42 patients (69.0%) showed moderate or high tumor-infiltrating CD8^+^ lymphocytes (inflamed). In the non-responsive group (stable disease and progressive disease), only 17 of the 42 patients (40.5%) showed moderate or high intratumoral CD8^+^ lymphocytes (*p* < 0.05). (**B**) The average number of tumor-infiltrating lymphocytes (TIL) in clinically responsive patients was significantly higher than in patients with disease progression (*p* < 0.05). There was no significant difference in TILs between the SD and PD groups. (**C**) In the responsive group, 13 of the 16 patients showed a high TIL/TC ratio (≥1). In the progress disease group, only 2 of the 6 patients showed a high TIL/TC ratio (*p* < 0.05). High: >1 TIL/TC; Low: <1 TIL/TC.

**Figure 3 cancers-16-00744-f003:**
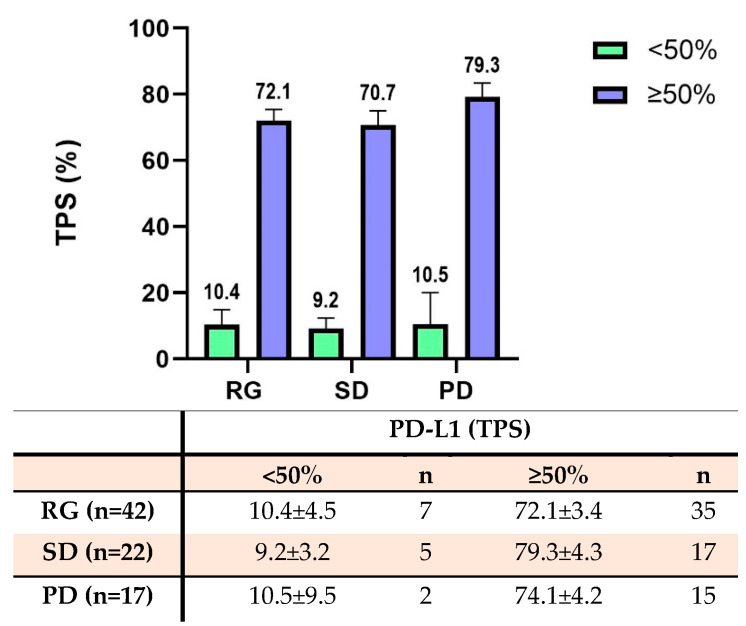
The expression of PD-L1 for NSCLC tumor cells according to the tumor proportion score (TPS). The bar diagram shows the PD-L1 TPS score across multiple groups. The table below summarizes the PD-L1 TPS score and patient number across multiple groups. There were no significant differences in the PD-L1 expression on tumor cells among the three treatment groups (*p* > 0.05).

**Figure 4 cancers-16-00744-f004:**
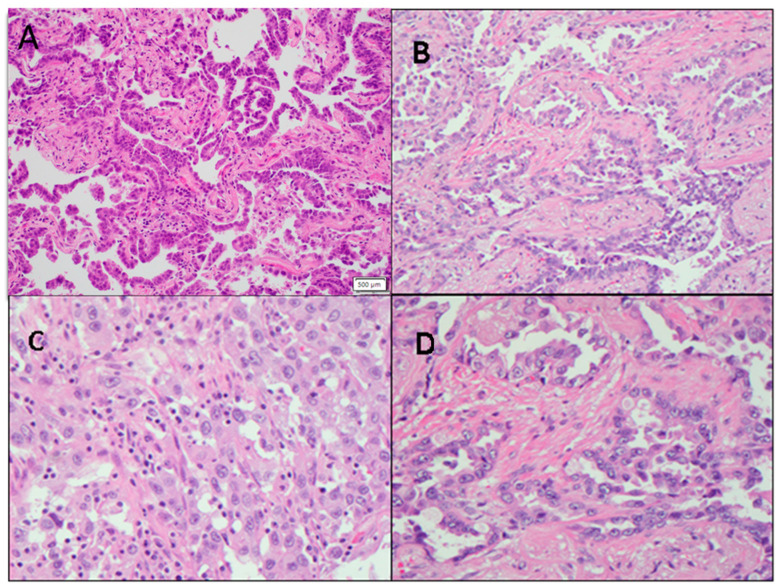
Representative non-small cell lung cancer images show variations in tumor cellularity and tumor-infiltrating lymphocytes (TILs). (**A**) NSCLC with high tumor cellularity (200×). (**B**) NSCLC with low tumor cellularity (200×). (**C**) NSCLC stroma with high TILs (400×). (**D**) NSCLC stroma with low TILs (400×).

**Figure 5 cancers-16-00744-f005:**
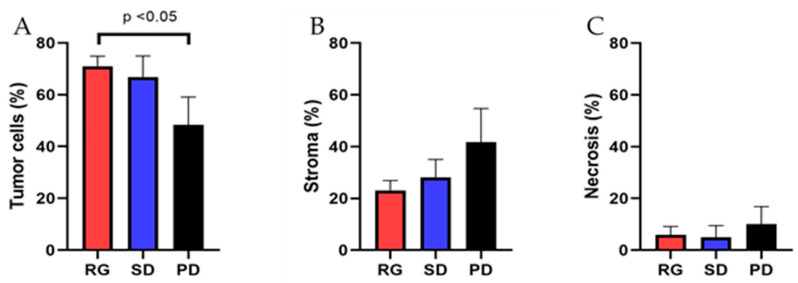
The association of the tumor cell content and therapeutic response in NSCLCs treated with Pembrolizumab. The parameters assessed include the proportion of viable tumor cells, extracellular stroma, and necrosis in tissue samples. (**A**) The proportion of tumor cells in the therapeutic response group (70.9 ± 4.0%) was significantly higher than that in the disease progression group (48.3 ± 10.8%). (**B**,**C**) There were no significant differences in the amount of extracellular stroma and tumor necrosis among the three groups.

**Table 1 cancers-16-00744-t001:** Clinicopathological characteristics.

Characteristic	Overall, n = 84 ^1^	Responsive, n = 42 ^1^	Stable, N = 23 ^1^	Progressive, n = 19 ^1^	*p*-Value ^2^
Age	65 (61, 74)	68 (62, 76)	65 (60, 71)	64 (57, 70)	0.2
Gender					0.6
Female	48 (57%)	23 (55%)	15 (65%)	10 (53%)	
Male	36 (43%)	19 (45%)	8 (35%)	9 (47%)	
Diagnosis					0.3
Adenocarcinoma	51 (61%)	26 (63%)	15 (65%)	10 (53%)	
Squamous cell carcinoma	18 (22%)	7 (17%)	6 (26%)	5 (26%)	
Adenosquamous carcinoma	1 (1.2%)	0 (0%)	1 (4.3%)	0 (0%)	
NSCLC	13 (16%)	8 (20%)	1 (4.3%)	4 (21%)	
Smoking					0.7
Current	27 (33%)	12 (30%)	9 (41%)	6 (32%)	
Former	46 (57%)	22 (55%)	12 (55%)	12 (63%)	
Never	8 (9.9%)	6 (15%)	1 (4.5%)	1 (5.3%)	
Second Malignancy	11 (13%)	6 (14%)	4 (17%)	1 (5.3%)	0.5

^1^ Median (IQR); n (%); ^2^ Kruskal–Wallis rank sum tests; Pearson’s Chi-squared test; Fisher’s exact test.

**Table 2 cancers-16-00744-t002:** Molecular alterations.

Group	Overall Mutation (n)	TP53 Mutation (n)	KRAS Mutation (n)
RG (n = 42)	40	24	19
SD (n = 23)	22	7	11
PD (n = 19)	17	11	8
Total (n = 84)	79	42	38

RG: Response group (complete response plus partial response); SD: Stable disease; PD: Progressive disease.

## Data Availability

The published data will be made available upon satisfactory written request.

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
