# Peer review of "Tumor Characteristics and Treatment Responsiveness in Pembrolizumab-Treated Non-Small Cell Lung Carcinoma"

_cancers, 2024, doi:10.3390/cancers16040744_

Round 1

Reviewer 1 Report

Comments and Suggestions for Authors

The authors sought to identify factors underlying differences in response to pembrolizumab between patient groups. Although some results were obtained, the reader failed to gain valuable information. The authors can then carefully consider the purpose of the study and refine the research strategy, including increasing the number of cases.

Here are some suggestions based on existing content:

1. Analysis of the significance of differences in clinical characteristics between different groups.

2. Evaluation criteria for CD8+ lymphocyte levels.

3. The Kaplan-Meiser survival curves of the 4 groups are in Figure 2A.

4. Lines 194-196, which groups of patients are clinically responding? In Figure 2B, p 0.05 for rg&pd and p0.05 for SD&PD. What p values?

5. Sequencing data of gene expression and mutations should be provided as attachments, and raw sequencing data should be uploaded to public databases.

6. Are lines 255-267 misplaced or not deleted?

​7. Reference 22 Analysis of differences between tumor-stromal ratios.

Author Response

Reviewer 1

Comments and Suggestions for Authors

The authors sought to identify factors underlying differences in response to pembrolizumab between patient groups. Although some results were obtained, the reader failed to gain valuable information. The authors can then carefully consider the purpose of the study and refine the research strategy, including increasing the number of cases.

  • We express our sincere gratitude to the reviewer for providing thorough and thoughtful comments. In the revised manuscript, we have made efforts to refine the purpose of the study. The data derived from this study emphasizes the significance of detailed pathological characteristics in predicting treatment responsiveness in pembrolizumab-treated NSCLC cases. However, we acknowledge that the relatively small number of cases is one of the limitations of our study. Future studies with a larger sample size will be necessary to validate our findings.

Here are some suggestions based on existing content:

  1. Analysis of the significance of differences in clinical characteristics between different groups.
  • We appreciate this suggestion. The clinical characteristics have been statistically analyzed and presented in Table 1. The revised table is shown below-

Table 1. Clinicopathologic characteristics

Characteristic

Overall, N = 841

Responsive, N = 421

Stable, N = 231

Progressive, N = 191

p-value2

age

65 (61, 74)

68 (62, 76)

65 (60, 71)

64 (57, 70)

0.2

gender

0.6

    Female

48 (57%)

23 (55%)

15 (65%)

10 (53%)

    Male

36 (43%)

19 (45%)

8 (35%)

9 (47%)

diagnosis

0.3

    Adenocarcinoma

51 (61%)

26 (63%)

15 (65%)

10 (53%)

    Squamous cell carcinoma

18 (22%)

7 (17%)

6 (26%)

5 (26%)

    Adenosquamous carcinoma

1 (1.2%)

0 (0%)

1 (4.3%)

0 (0%)

    NSCLC

13 (16%)

8 (20%)

1 (4.3%)

4 (21%)

smoking

0.7

    Current

27 (33%)

12 (30%)

9 (41%)

6 (32%)

    Former

46 (57%)

22 (55%)

12 (55%)

12 (63%)

    Never

8 (9.9%)

6 (15%)

1 (4.5%)

1 (5.3%)

secondMalignancy

11 (13%)

6 (14%)

4 (17%)

1 (5.3%)

0.5

1 Median (IQR); n (%)

2 Kruskal-Wallis rank sum test; Pearson’s Chi-squared test; Fisher’s exact test

  1. Evaluation criteria for CD8+ lymphocyte levels.
  • This has been clarified as follows: The levels for CD8 TILs were recorded from molecular profiling data (Omniseq Inc) from patients’ medical record. Per report, CD8+ lymphocytes were analyzed using RNA sequencing. Interpretation of gene expression for CD8 TIL is done as follows: Each gene is compared to a reference population derived from 735 unique tumors, normalized to a value between 1 and 100 and scored as the percentile (relative) rank (%rank). CD8 TILs gene expression is expressed as high for genes ranked 75-100 (highly inflamed), moderate for genes ranked 25-74 (moderate inflamed), and low for genes ranked 0-24 (non-inflamed). In our analyses, the tumor was categorized as inflamed if the CD8 gene expression profile was high or moderate and non-inflamed if the expression profile was reported as low.
  1. Lines 194-196, which groups of patients are clinically responding? In Figure 2B, p 0.05 for rg&pd and p0.05 for SD&PD. What p values?
  • We apologize for this confusion. This statement has been revised as follows-

In the treatment-responsive group (complete and partial response), 29 of 42 patients (69.0%) showed tumor infiltrating CD8+ lymphocytes. In the non-responsive group (stable disease and progressive disease), only 17 of 42 patients (40.5%) showed intratumoral CD8+ lymphocytes (p <0.05). There was no significant difference between SD and PD groups.

  1. Sequencing data of gene expression and mutations should be provided as attachments, and raw sequencing data should be uploaded to public databases.
  • Summary of mutation profile is provided in the supplemental table. Unfortunately, we don’t have access to raw sequencing data currently.
  •  

Group

Mutation_TP53

Mutation_KRAS

Mutation_Other

RG

C238F

NOTCH1

P278R, H179Q

CDKN2A, RAF1, TERT PROMOTER SNV, STK11, ATM

D148Y

NF1, APC, PIK3R1

ATM

C733G

G12V

BRAF, AR E710D

E271Q

G12V

STK11

e343x

g13d

atm, gata3, kit

BRAF, PIK3R1

Y205f

CDKN2A, ATM,  APC

G12S

BRCA2,

A159P, R175H

CDKN2A, NF2, APC, CDH1, PIK3R1, SMAD4, FBXW7

G12C

E286A, R306

Q61L

BRAF, IDH, MARP2K, PIK3CA,

R337L

G12C

CDKN2A, VHL

G12C

ABL1

G12C

TSC2

H193L

PIK3R1

NRAS, BRAF

g262v

CCND1, PIK3CA, CDKN2A, APC, CD274, PDCD1LG2, PIK3R1

G12C

CTNNB1, BRCA1, PIK3CA

T123M

Q61H

K132N

KIF5Be15-RETe12, CDKN2A

G266X,

NF1,TSC2, RET

G12C

BRAF, ATM, SMAD4

Y236C

NFE2L2, PTEN, CDKN2A

C135F

TSC2, FGFR3e17-TACC3e10, RB1

FBXW7

G12A

NF1, IFITM3

E171X

TSC2, STK11

r337l

BRCA2, NRAS, APC, NF2, PIK3R1, TET2

C12A

Y

G12C

Splice site SNV

BRAF(V600E), SMAD4

CDKN2A,

G12C

CDKN2A,

R273L

MET, NRAS, ROS1

V173G

A159P

BRCA2, EGFR, NFE2LE, pik3r1, ATM, TERT2

EGFR SLICE SITE SNV

G12A

ATM, PIK3CA

SD

C445del

R175H

CDKN2A, PIK3CA, ATM

BRCA2

G12C

g12v

EGFR, CDKN2A, NET, CDK6, BRAF EGFR

R249S

G12C

G13D

NRS, KEAP1, STK11

G12F

BRCA2, ATM

R273L

G12V

BAP1

NF1

G12C

G12V

G245V

STK11,FGFR2, APC, PDGFRA

PIK3CA

FGFR2, EZH2, TERT PROMOTER SNV, NOTCH1

Q61R

APC

E349fs

N131Y, G154fs

G12C

TSC1, KIT , NTRK1. AR, ATM, KIT AMPLIFICATION

BRCA2

PD

G13D

EGFR

S166X

EGFR, CTNNB1

G12V

APC, NF1, PDCD1LG2

g12c

CDKN2A, TET2

L114FfsX35

GATA3

G12V

G12C

ATM, VHL

c517g

PTEN, FGFR2

G266R

DDR2, CDKN2A, GATA3, STK11

H193L

ATM, MARPK1

G12C

EGFR, EML4-ALK, ARID1A

V157F

G12C

APC

E204X

R273H, P142F

NRAS

G12V

  •  
  1. Are lines 255-267 misplaced or not deleted?
  • Our apology for this editorial error. This has been corrected.

​6. Reference 22 Analysis of differences between tumor-stromal ratios.

We agree.

Reviewer 2 Report

Comments and Suggestions for Authors

The study seems to be excellent. The only issue is that material gained from biopsy and surgery is not comparable. First of all You do not mention whether it was fine needle biopsy or core biopsy. Secondly material resected during surgical procedure differs in pathological microscopic view as a lot of different processes ( including inflammation and lymphocyte migration) is generated also by surgical procedure itself. It will be marvelous to continue this study and to make patients group more uniform, thanks to analysis of   tissue samples acquired exclusively surgically. 

Author Response

Reviewer 2

The study seems to be excellent. The only issue is that material gained from biopsy and surgery is not comparable. First of all, you do not mention whether it was fine needle biopsy or core biopsy. Secondly material resected during surgical procedure differs in pathological microscopic view as a lot of different processes (including inflammation and lymphocyte migration) is generated also by surgical procedure itself. It will be marvelous to continue this study and to make patients group more uniform, thanks to analysis of   tissue samples acquired exclusively surgically. 

  • We greatly appreciate these comments. Regarding the biopsy samples, the tissue was collected through core needle biopsies. While the sample comprised both biopsies and resection specimens, the majority consisted of biopsy tissue, given that patients with advanced disease stage are rarely treated with surgical resection of the tumor. It would be meaningful to have a more uniform population of samples involving surgically resected tissue when a reasonable sample size becomes available in the future. We acknowledge the fact that surgical resection can initiate tissue inflammation, often involving neutrophils rather than lymphocytes. These limitations are discussed in the revised manuscript.

Round 2

Reviewer 1 Report

Comments and Suggestions for Authors

The author made modifications based on the review comments and has no further suggestions.